# The Relationship Between Systemic Immune Inflammatory Level and Dry Eye in Patients with Sjögren’s Syndrome

**DOI:** 10.3390/jcm13226840

**Published:** 2024-11-14

**Authors:** Ülkem Şen Uzeli, Ayşe Gülşen Doğan, Tayfun Şahin

**Affiliations:** 1Internal Medicine Department, Hitit University, Corum 19030, Turkey; 2Department of Physical Medicine and Rehabilitation, Hitit University, Corum 19030, Turkey; drmdagu@gmail.com; 3Department of Ophthalmology, Hitit University, Corum 19030, Turkey; drtayfunsahin82@gmail.com

**Keywords:** Sjögren Syndrome, dry eye, systemic immune-inflammation index

## Abstract

**Background and objectives**: Sjögren Syndrome (SS) is a chronic, systemic, and progressive autoimmune disease in which inflammatory processes play a role. Dry eyes or mouth are present in approximately 95–98% of patients with pSS. This study aimed to evaluate the relationship between SII level and disease activity as well as dry eye involvement in patients with pSS. **Materials and methods**: A cross-sectional design was employed, and a total of 28 female patients who were aged 18–65 years and were diagnosed with pSS were involved. The Sjögren Syndrome Disease Activity Index (ESSDAI) was calculated in patients. The Schirmer test was applied to all patients. The relationship between SII level and disease activity as well as dry eye involvement in pSS patients was evaluated. **Results**: In our study, a strong positive correlation was found between the SII value and pSS disease activity, while a negative correlation was found between the Schirmer test, which shows dry eye findings, and eye drying time, and a positive correlation was found with the OSDI. **Conclusions**: this study reported a correlation between hematological parameters and the development of dry eye in pSS. NLR, PLR, and SII showed statistically significant changes in pSS patients.

## 1. Introduction

Sjögren Syndrome (SS) is a chronic, systemic, and progressive autoimmune disease in which inflammatory processes play a role [1]. It is closely related to other connective tissue diseases and can be seen as part of them [2]. The concepts of primary and secondary SS have been used since 1965, and primary Sjögren Syndrome (pSS) is distinguished from secondary SS by the fact that it is not accompanied by autoimmune diseases, such as rheumatoid arthritis, systemic lupus erythematosus, and systemic sclerosis [3,4]. The differential diagnosis of these two clinical conditions is very critical in determining the treatment, follow-up principles, and prognosis of the disease [5]. Dry eyes or mouth are present in approximately 95–98% of patients with pSS, and both are seen in 89% of patients [6]. In this disease, where the lymphocytic infiltration of exocrine glands is the main histopathological indicator, the decrease in tear secretion and the change in its content as a result of the lymphocytic infiltration of lacrimal glands cause xerophthalmia and the accompanying symptoms in patients. Patients present with symptoms, such as redness in the eye, constant itching, a sensation of sand being thrown into the eye, sensitivity to light, a foreign-body sensation in the eye, and decreased tear production [7]. Autoantibodies, as in all autoimmune diseases, play a notable role in the diagnosis of pSS and the evaluation of clinical properties. pSS cases usually test positive for ANA, anti-Ro, anti-La, and RF, and anti-Ro and anti-La are pSS-specific antibodies [8]. In recent studies, the platelet–lymphocyte ratio (PLR) and neutrophil–lymphocyte ratio (NLR) have been found to be associated with inflammatory diseases [9,10]. The systemic immune-inflammation index (SII), calculated by using platelet, neutrophil, and lymphocyte counts together, is a much more specific marker for showing inflammation and immune response compared to the PLR and NLR. Studies have shown that high SII values are associated with the severity of the disease and poor prognosis in many diseases and malignancies [11,12]. We did not come across any study in the literature on SII in patients with pSS. This study aimed to evaluate the relationship between SII level and disease activity as well as dry eye involvement in patients with pSS.

## 2. Materials and Methods

A cross-sectional design was employed, and a total of 28 female patients who were aged 18–65 years presented to the Hitit University Erol Olçok Training and Research Hospital Department of Internal Medicine between June and July 2024, and were diagnosed with pSS according to the 2016 ACR/EULAR classification criteria, were included in the study. The approval of the Hitit University Clinical Research Ethics Committee was obtained for the study (date: 5 June 2024, Decision No: 2024-34). Well-informed written consent was obtained from all participants according to the principles of the Declaration of Helsinki. The control group consisted of 28 healthy women with demographic characteristics similar to the patient group in terms of age. Inclusion criteria included having no other systemic disease and history of contact lens use, and not being in the pregnancy lactation period. The demographic data of all participants were recorded. The EULAR Sjögren Syndrome Disease Activity Index (ESSDAI), which determines the disease activity index, was filled out for the patient group. The ESSDAI is used to assess twelve features: constitutional symptoms (fever, night sweats, weight loss), lymphadenopathy and/or lymphoma, and glandular (swelling of salivary and/or lacrimal glands), joint, cutaneous, pulmonary, renal, muscular, CNS, peripheral nervous system (PNS), hematological, and biological (complement, cryoglobulins, hypergammaglobulinemia) domains. Each feature is given a value between 0 and 3 depending on the activity level and between 1 and 6 depending on the weight of each feature. These two values are multiplied by 19 for each feature. Finally, the activity index is calculated by summing the values obtained for each domain. The maximum ESSDAI score is theoretically 123. However, these scores are lower in real patients [13,14]. The ocular surface disease index (OSDI) questionnaire was applied to all patients during the examination to evaluate the stages of dry eye disease and their effects on patients’ daily living. The OSDI questionnaire is the most widely used in dry eye disease studies and consists of 12 questions. This questionnaire is used to evaluate the symptoms of ocular irritation associated with dry eye and the effects of these symptoms on visual functions. Since the frequency of symptoms is also questioned, the severity of dry eye can also be graded. Each question is scored between 0 and 4. The OSDI score ranges from 0 to 100. Scores of ≥13 on the OSDI questionnaire are in favor of dry eye [15].

During the clinical evaluation, a detailed anterior segment examination was performed to determine the dry eye symptoms and their severity. In this examination, invasive tear break-up time was measured using fluorescein sodium [16]. To do this, the patient is asked to blink 3–4 times. The time from the patient’s last blink to the formation of a dry area on the corneal surface measured under blue cobalt light on the biomicroscope is recorded. The average of 3 measurements is taken, and if the average of these measurements is below 10 s, it shows decreased tear stability [17]. A Schirmer test, applied without the use of topical anesthetic, is performed. It is performed by placing filter papers of 5 × 35 mm in the area where the middle 1/3 and the outer 1/3 of the lower eyelid meet in the lower conjunctival fornix, and the amount of wetness on the paper is noted after five minutes. The Schirmer 1 test performed without anesthesia is used to measure reflex tear secretion together with basal tear secretion. A measurement below 10 mm after five minutes is significant for dry eye [18]. The SII level obtained from the hemogram parameter of all patients and healthy groups was calculated using the formula of platelet count × neutrophil count/lymphocyte count. A high SII value indicates relatively increased neutrophil and platelet counts and low lymphocyte counts, which is a sign of a strong inflammatory response [19].

Data were evaluated on the IBM SPSS Statistics Standard Concurrent User V 29 (IBM Corp., Armonk, NY, USA) statistical software package. Descriptive statistics were presented using the number of units (*n*), percentages (%), mean ± standard deviation, geometric mean ± standard error, median, and interquartile range values. The normality of quantitative data was assessed using the Shapiro–Wilk test. The variance homogeneity of the groups was analyzed with the Levene test. Two-group comparisons for numerical variables were made using the independent sample *t*-test for data with a normal distribution, and the Mann–Whitney U test for non-normal distributions. The relationship between SII values and numerical variables was evaluated using the Spearman correlation coefficient. Variables affecting SII were evaluated using multiple linear regression analysis. The backward elimination method was employed to reach the final model. A *p* value < 0.05 was considered statistically significant.

## 3. Results

According to Table 1, there was no statistically significant difference between the age, hemoglobin, and monocyte values of the control and patient groups. The neutrophil values of the patient group were statistically higher than those of the control group. The patient group had statistically lower lymphocyte values than the control group. The platelet, NLR, PLR, and SII values of the patient group were statistically higher than those of the control group.

According to Table 2, there was no statistically significant relationship between SII values and age. SII values had a statistically significant, moderate-level negative correlation with tear break up time right and left values. They had a statistically significant and good negative correlation with Schirmer right and left values. There was a statistically significant and strong positive correlation between SII and ESDAII values. Also, a statistically significant and good-level positive correlation was determined between SII and OSDI values.

Table 3 presents the evaluation of factors affecting SII using multiple linear regression analysis. In the first stage, variables in Table 2 (tear break up time left, Schirmer left, ESDAII, and OSDI) were included in the model. Since there was a high correlation between tear break up time left and tear break up time right (*rho* = 0.900; *p* < 0.001), only the tear break up time left value was included in the model. As Schirmer right and Schirmer left values were the same, Schirmer left value was included in the model. ESDAII was included in the final model. According to the results in Table 3, a one-unit increase in the ESDAII score caused a 25.7 increase in SII. The established model was statistically significant and it met the assumptions of linear regression analysis.

## 4. Discussion

The results of our study showed that neutrophil, platelet, NLR, PLR, and SII levels were higher in patients with pSS, while the lymphocyte count was low. A strong positive correlation was found between the SII value and disease activity. The Schirmer test, which shows dry eye findings, was correlated negatively with tear break up time and positively with OSDI.

The role of hematological parameters in the evaluation and follow-up of autoimmune disease activity has increased significantly recently. In particular, lymphocytes, monocytes, neutrophils, and platelets, as measured from hemogram parameters, play notable roles in inflammatory processes [20]. NLR and PLR obtained from these cell counts are considered systemic parameters of inflammation and have been associated with increased disease activity in autoimmune disorders, such as systemic lupus erythematosus, rheumatoid arthritis, and ankylosing spondylitis [21,22,23]. In the study by Yıldız et al. on 41 patients with pSS, NLR and PLR levels were found to be significantly higher in the pSS group than in the healthy group and a positive relationship was found between PLR and ESSDAI. In the study by Mihai et al., NLR and PLR were found to be significantly higher in 124 patients with pSS than in the control group [20]. In our study, similar to the literature, PLR and NLR levels were found to be significantly higher in patients with pSS than in the healthy group.

SII is a biomarker obtained from hemogram parameters that are significant in terms of disease activity and prognosis in patients with autoinflammatory and cardiovascular diseases, and malignancies. In studies including patients with psoriatic arthritis, SII levels were found to be significantly higher in patient groups than in the healthy group and significantly positively correlated with disease activity [24,25]. Similarly, it was found that SII levels were significantly higher in patients with rheumatoid arthritis compared to healthy controls and that SII was associated with the severity of the disease. These findings suggested that SII may have been a useful biomarker for monitoring inflammation and disease activity in patients with RA [26,27]. We did not find any study in the literature on the examination of SII levels in patients with pSS. In our study, SII levels were found to be significantly higher in the patient group than in the healthy group. In addition, when the relationship with disease activity was examined, a significant relationship was found between ESSDAI and SII. This showed us that SII could be used as an inflammation marker in disease activity in pSS.

pSS is a systemic autoimmune disease mainly affecting exocrine glands such as salivary and lacrimal glands. It leads to inflammation that can progress from asymptomatic conditions to systemic complications and lymphoma development [28]. Many patients with pSS experience extra-glandular symptoms, such as cardiac, joint, gastroenterological, skin, pulmonary, renal, and neurological involvements [4]. Inflammation in the body has been blamed as the underlying cause of all these systemic complications. The identification of easily accessible disease predictive parameters in pSS may prevent possible complications and lead to more effective treatment in pSS. In a study by Mihai et al. involving 121 patients with pSS, increased NLR values were associated with higher ESSDAI scores, which were positively correlated with neurological involvement [20]. In a study by Zhang et al., it was shown that NLR could be used to predict interstitial lung disease in pSS and that high NLR values were associated with poor prognosis [29]. In another study by Mihai et al. on the investigation of the development of vasculitis in patients with pSS, it was determined that NLR and PLR levels showed a significant positive correlation in the development of vasculitis [30]. In our study, SII values showed a significant negative correlation with the tear break up time and Schirmer test, while a significant positive correlation was found with OSDI. Our results suggested that high SII values could be used to predict dry eye development.

Our study has some limitations. For example, the number of patients was small, the duration of dry eye symptoms was not questioned, and the control group did not include patients who only had only dry eye disease, without additional diseases. Multicenter, prospective studies on larger patient cohorts are necessary to further confirm our findings regarding the predictive role of hematological parameters in ocular involvement in patients with pSS and to improve the optimized cut-off values for the parameters that were evaluated. To our knowledge, this is the first study to evaluate SII levels and examine their relationship with disease activity and dry eye findings in patients with pSS. We think that this study will make valuable contributions to the literature by elucidating the pathogenesis of pSS and making an earlier diagnosis of dry eye development.

## 5. Conclusions

This study reported a positive correlation between hematological parameters and the development of dry eye in pSS. NLR, PLR, and SII were significantly higher in pSS patients. Our results suggest that these cost-effective, reliable, and widely available biomarkers may be potentially useful elements for the early detection of patients at risk for dry eye manifestations. Furthermore, these biological parameters may be useful biomarkers for clinicians to monitor disease progression and identify potentially severe glandular manifestations in patients with pSS. Early diagnosis and appropriate treatment will prevent the development of many complications associated with pSS. Early diagnosis and treatment will prevent the development of chronic xerophthalmia and xerostomia-related damage.

## Figures and Tables

**Table 1 jcm-13-06840-t001:** Comparison of age and hemogram parameters between groups.

	Groups		Test Statistics	
	Control *n* = 28	Patient *n* = 28	Test Value	*p* Value
**Age**	50.7 ± 8.5	51.4 ± 10.1	0.272	0.787 ^†^
**HB**	13.22 ± 1.25	13.35 ± 1.25	−0.395	0.695 ^†^
**Neutrophil**	3.01 ± 0.79	4.81 ± 0.90	7.906	**<0.001** ^†^
**Lymphocyte**	2.30 ± 0.56	1.43 ± 0.42	6.628	**<0.001** ^†^
**Monocyte**	0.435 (0.178)	0.410 (0.278)	0.541	0.588 ^&^
**Platelet**	238.5 (74.5)	419.5 (62.0)	6.179	**<0.001** ^&^
**NLR**	1.41 ± 0.64	3.56 ± 0.91	10.191	**<0.001** ^†^
**PLR**	108.8 (24.6)	285.6 (113.3)	6.129	**<0.001** ^&^
**SII ***	326.1 ± 140.8	1415.8 ± 385.1	17.637	**<0.001** ^†^

Summary statistics were given as mean ± standard deviation or median (interquartile range) values. * Comparisons were made on log base 10-transformed data, and summary statistics were given as geometric mean ± standard error of true values. ^†^: Independent sample *t*-test; ^&^: Mann–Whitney U test.

**Table 2 jcm-13-06840-t002:** Relationship of SII values with age, eye findings, ESDAII, and OSDI.

	SII	
	*rho*	*p*
**Age**	0.238	0.222
**DET Right**	**−0.431**	**0.022**
**DET Left**	**−0.457**	**0.014**
**Schirmer Right**	**−0.658**	**<0.001**
**Schirmer Left**	**−0.658**	**<0.001**
**ESDAII**	**0.818**	**<0.001**
**OSDI**	**0.627**	**<0.001**

*rho*: Spearman correlation coefficient.

**Table 3 jcm-13-06840-t003:** Evaluation of factors affecting SII using multiple linear regression analysis.

	Regression Coefficients						
	*β*	*sh*	*zβ*	*t*	*p*	95.0% Confidence Interval for *β*	
						Lower Limit	Upper Limit
**Constant**	1190.7	47.7		24.959	<0.001	1092.7	1288.8
**ESDAII**	25.7	2.9	0.869	8.944	<0.001	19.9	31.7
*β*: Regression coefficient, *sh*: Standard error, *zβ*: Standardized regression coefficient							
Variables included in the model: Tear break up time left, Schirmer left, ESDAII, OSDI							
Elimination method: Backward							
Model statistics: *F* = 80.003; *p* < 0.001, *R*^2^ = 0.755, Corrected *R*^2^ = 0.745							
Shapiro–Wilk test statistics for normality of standardized error terms: Statistics = 0.938; *p* = 0.100							
Autocorrelation between error terms: Durbin-Watson = 2.189							

## Data Availability

The datasets used and/or analyzed during the current study are available from the corresponding authors upon reasonable request.

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
