# Peer review of "The Relationship Between Systemic Immune Inflammatory Level and Dry Eye in Patients with Sjögren’s Syndrome"

_jcm, 2024, doi:10.3390/jcm13226840_

Round 1
Reviewer 1 Report
Comments and Suggestions for Authors
The manuscript by Ülkem ŞEN UZELİ et al discusses the relationship between systemic immune-inflammatory levels and dry eye in patients with Sjögren's Syndrome, focusing on the use of the Systemic Immune-Inflammation Index (SII) as a marker for disease activity and prognosis. This research adds valuable information to the field and highlights the potential of SII as a marker for disease activity and prognosis in pSS patients.
In this study, the authors have investigated the relationship between systemic immune inflammation levels and dry eye in patients with primary Sjögren's syndrome (pSS). The important points include Sjögren's Syndrome (SS): A chronic autoimmune disease affecting exocrine glands, leading to symptoms like dry eyes and mouth. Study Design: Cross-sectional study with 28 female pSS patients and 28 healthy controls. The Key Findings are: pSS patients had higher neutrophil, platelet, NLR, PLR, and SII levels, and lower lymphocyte counts compared to controls. SII showed a strong positive correlation with disease activity (ESSDAI) and dry eye severity (OSDI), and a negative correlation with Schirmer test results. The overall Implications are: SII can be a useful biomarker for monitoring inflammation, and disease activity, and predicting dry eye development in pSS patients.
The major limitation of the study is the small sample size and lack of a control group with only dry eye disease.
Overall, the study suggests that hematological parameters like NLR, PLR, and SII are cost-effective and reliable biomarkers for early detection and monitoring of pSS.
Other important limitations
The abstract body is missing from the pdf proof provided
The author list and affiliations are all missing or not provided in the pdf document.
Authors conflict of interest statement is missing
The Ethics committee approval number is missing - The approval of the Hitit University Clinical Research Ethics Committee was obtained for 40 the study (Date: xxx, Decision No: xx).
Highly recommend the manuscript to be proofread as per journal guideline
Author Response
Dear Referee
1) Ethics committee decision number added.
2) Abstract added to text file.
3) Authors' conflict of interest declaration added.
4) Author list and institutions they are affiliated with rearranged in pdf document.
King Regards
Reviewer 2 Report
Comments and Suggestions for Authors
This manuscript described a study to explore correlations of some hematological parameters with primary Sjogren Syndrome and correlations of Systemic Immune-Inflammation with this condition. The introduction has provided sufficient background information regarding this condition and clearly stated the aim of the study. The method part included detailed description of the work carried out. The result part is concise, and well structured and presented. The discussion is adequate with two limitations of this study mentioned at the end. However, I have a few comments to make:
1) The abstract and author information seem to be missing for this version.
2) The authors could consider to capitalize S of syndrome in line 8 and I of immune-inflammation in line 27.
3) Please clarify what the Date and Decision No. are in line 41.
4) The authors may need to combine statistical methods into method part.
5) Please add a proper tile for Table 1
6) What is BKT in line 129?
7) Since one of the benefits from the findings in this study is for early detection, so please write about the potential consequences if it is not detected early.
8) In conclusion, the author need to specify the findings rather than just say there is a correlation. Also, the author could consider use full name rather than acronyms in conclusion so that the reader can understand them without reading the whole article.
Author Response
Dear Referee
1) Abstract section added.
2) Authors capitalized the letter S of syndrome in line 8 and the letter I of immune-inflammation in line 27.
3) Date and Decision Number in line 41 were specified.
4) Statistical methods were combined in the method section.
5) Please add a suitable tile for Table 1. (Comparison of age and hemogram parameters between groups)
6) Tear break-up time took its place in the entire text with its long version.
7) Early diagnosis and appropriate treatment will prevent the development of many complications associated with Pss. Early diagnosis and treatment will prevent the development of chronic xerophthalmia and xerostomia-related damage.
8) As a result, it was enriched with findings.
Kind Regards
Round 2
Reviewer 1 Report
Comments and Suggestions for Authors
Thank you for addressing my comments.
Reviewer 2 Report
Comments and Suggestions for Authors
Thank you for addressing my comments.